# Bacteria and Host Interplay in *Staphylococcus aureus* Septic Arthritis and Sepsis

**DOI:** 10.3390/pathogens10020158

**Published:** 2021-02-03

**Authors:** Tao Jin, Majd Mohammad, Rille Pullerits, Abukar Ali

**Affiliations:** 1Department of Rheumatology and Inflammation Research, Institute of Medicine, Sahlgrenska Academy, University of Gothenburg, 413 46 Gothenburg, Sweden; majd.mohammad@rheuma.gu.se (M.M.); rille.pullerits@rheuma.gu.se (R.P.); abukar.ahmed.ali@rheuma.gu.se (A.A.); 2Department of Rheumatology, Sahlgrenska University Hospital, 413 45 Gothenburg, Sweden; 3Department of Clinical Immunology and Transfusion Medicine, Sahlgrenska University Hospital, 413 45 Gothenburg, Sweden

**Keywords:** *Staphylococcus aureus*, septic arthritis, sepsis, virulence factors, immunity

## Abstract

*Staphylococcus aureus* (*S. aureus*) infections are a major healthcare challenge and new treatment alternatives are needed. *S. aureus* septic arthritis, a debilitating joint disease, causes permanent joint dysfunction in almost 50% of the patients. *S. aureus* bacteremia is associated with higher mortalities than bacteremia caused by most other microbes and can develop to severe sepsis and death. The key to new therapies is understanding the interplay between bacterial virulence factors and host immune response, which decides the disease outcome. *S. aureus* produces numerous virulence factors that facilitate bacterial dissemination, invasion into joint cavity, and cause septic arthritis. Monocytes, activated by several components of *S. aureus* such as lipoproteins, are responsible for bone destructions. In *S. aureus* sepsis, cytokine storm induced by *S. aureus* components leads to the hyperinflammatory status, DIC, multiple organ failure, and later death. The immune suppressive therapies at the very early time point might be protective. However, the timing of treatment is crucial, as late treatment may aggravate the immune paralysis and lead to uncontrolled infection and death.

## 1. *S. aureus* and Its Virulence Factors

Nearly half of the human population is at some point colonized by *Staphylococcus aureus (S. aureus)*. Of these, 20% are persistently colonized while around 30% are intermittently colonized, mostly in the anterior nares and the skin [1]. However, one should not make the mistake of assuming that *S. aureus* is a harmless microbe that is only part of the normal flora. Rather, *S. aureus* is indeed a very virulent bacteria that causes a wide range of diseases, from simple wound infections and food poisoning to life-threatening conditions such as sepsis, meningitis, and endocarditis [2]. In this review, we focus on septic arthritis and sepsis, which are two of the many severe infections caused by *S. aureus*.

*S. aureus* is a very resilient pathogen due to the various virulence factors it contains and produces, some of which are described below and illustrated in Figure 1. The biological function of *S. aureus* virulence factors and their roles in infections are summarized in Table 1, reviewed in [3].

### 1.1. Cell Wall Components

*S. aureus* expresses a capsular polysaccharide (CP) that functions as a virulent factor, enabling the bacteria to evade phagocytosis [4]. Several serotypes of the CP have been identified and of those, CP5 and CP8 are the major ones. Most of the clinical isolates of *S. aureus* have the capability to produce either CP5 or CP8 [4]. *S. aureus* strains expressing the CP5 capsule significantly increase mortality and arthritis frequency and severity in *S. aureus* induced sepsis and septic arthritis, respectively, compared to the strains lacking the CP5 capsule [5]. This can be due to the downregulatory effects of CP5 on the uptake and intracellular killing ability of the phagocytes [5]. The CP8 serotype seems to be less virulent than the CP5 serotype as demonstrated by the ability of CP5 to cause higher bacteremia than the CP8 serotype in a mouse model of bacteremia. The CP5 producing strain also exhibited greater resistance to in vitro opsonophagocytic killing by neutrophils compared to the CP8 serotype [6].

The cell wall of *S. aureus* is made up of a 20–30 nm thick layer of peptidoglycan (PGN). Apart from being a protective barrier of the bacteria, PGN has other functions such as being a scaffold, whereby surface proteins that are fundamental for bacterial virulence can attach [7]. The major structural features of PGN consist of linear glycan strands made up of alternating N-acetylglucosamine and N-acetylmuramic acid residues that are linked by β-1-4 bonds [8]. The glycan strands are cross-linked by short peptides made up of D-alanine, L-lysine, D-glutamic acid, and L-alanine [9]. The ɛ-amino groups L-lysine of nearby peptides are cross-linked to D-alanine of other peptides through pentaglycine bridges, thus giving rise to the 3-dimensional structure of the PGN [10]. Due to the critical role it plays in maintaining bacterial structure, growth, and viability, PGN is a target for antibiotics and for the immune system. Nod-like receptors (NLRs), mannose-binding lectin, and lysozyme are some of the components of the immune system that can recognize and target PGN [11]. PGN is a strong inducer of inflammation and stimulates the release of proinflammatory cytokines such as tumor necrosis factor alpha (TNF-α), interleukin (IL)-1β, and IL-6. Furthermore, studies have shown that PGN alone can induce arthritis in mice [12] and repetitive inhalation of PGN components can lead to bone loss [13]. However, compared to an extremely potent effect of *S. aureus* lipoproteins (Lpps) the proinflammatory capacity of PGN is mild and transient [14].

Besides, secondary modifications of the PGN cell wall are of vital importance in order to resist host immune response enforcements [15]. Antibacterial enzymes produced by the host, such as lysozyme, also known as muramidase, can cleave PGN in the β-1,4 linkage site localized between the sugar residues of N-acetylglucosamine and N-acetylmuramic acid, thereby inhibiting bacterial overgrowth [15]. However, *S. aureus* impressively protects itself from “cell wall breakdowns” by utilizing *O*-acetylation modifications mediated by the *O*-acetyltransferase A (*OatA*) enzyme in the PGN cell wall [16]. We demonstrated that the virulence associated with PGN *OatA,* in both systemic and local *S. aureus*-induced septic arthritis model, led to milder progression of the disease in mice infected with a Δ*oatA* mutant strain [17], indicating the pathogenic importance of the *O*-acetylation of PGN in staphylococcal septic arthritis.

Among the constituents of the cell wall of *S. aureus* are the teichoic acids, made up of polyribitol and polyglycerol phosphates and sugar-containing polymers [18,19]. Teichoic acids are not unique to *S. aureus* but are found in various Gram-positive bacteria. They can be categorized into wall teichoic acids (WTAs) that are covalently attached to the bacterial PGN, and lipoteichoic acids (LTAs) that are anchored to the lipid membrane through a glycolipid [18,19]. Historically, LTA has been regarded as a potent stimulator of the innate immunity upon its recognition by Toll like receptor 2 (TLR2) among others, leading to the activation of the macrophages and the release of proinflammatory cytokines [20,21]. However, recent studies cast a shadow of doubt on these results, pointing a finger to contamination by *S. aureus* Lpps.

*S. aureus* Lpp consists of a lipid-moiety and a protein-moiety. The lipid portion is covalently attached to a cysteine residue in the N-terminal region, ultimately facilitating its anchoring in the outer leaflet of the bacterial cytoplasmic membrane [22]. Lpp in *S. aureus* plays an essential role in enabling the bacteria to acquire sufficient loads of iron under infectious conditions. As free iron ions are limited in the host environment and the iron supply is of critical importance for the survival of the staphylococcal pathogen [23,24], inhibition of the Lpp function, through incomplete Lpp maturation, can be deleterious to the metabolic fitness of the bacterium, consequently prompting it to cease in the battle against the host. The lipid structures of Lpp in *S. aureus* are known as microbe-associated molecular patterns (MAMPs) [22], being vital activators of the innate immunity as they play a potent role in alerting TLR2 in host cells [25].

*S. aureus* Lpp behave differently in different murine models, depending on the route of infection, the organ of interest, and the assessed time points. In a local knee arthritis model, *S. aureus* Lpps mediated severe arthritogenic and bone destructive effects in NMRI and C57BL/6 wild-type mice, but not in TLR2 deficient mice after intra-articular knee joint challenge with the purified Lpp. The arthritogenic properties were mediated through the lipid-moiety [14]. Interestingly, *S. aureus* SA113 Δ*lgt* mutant strain that lacks Lpp maturation, induced more knee swelling compared to its parental strain. This coincided with increased IL-6 expression and higher bacterial burden in the local knee [14]. In contrast, in a hematogenous septic arthritis model, increased bacterial persistence was observed in both C57BL/6 wild-type and TLR2 deficient murine kidneys when inoculated intravenously with the *S. aureus* Newman parental strain compared to its Δ*lgt* mutant strain [26]. This is in agreement with earlier reports showing that different organs, including the kidneys, exhibited higher bacterial loads when infected with SA113 parental strain in comparison to the Δ*lgt* mutant strain, independent of TLR2 and MyD88 signaling [27]. In addition, USA300 MRSA parental strain led to higher bacterial burden in kidneys of Balb/c mice compared to its Δ*lpl* mutant strain, lacking the *lpl* gene cluster [28].

### 1.2. Surface Proteins

*S. aureus* expresses several surface proteins that play a crucial role in enabling the bacteria to adhere to the host cells, aid in invasion of the bacteria, and evade the immune response mounted by the host [29]. Adherence of bacterial products to host tissues is one of the important steps in initiation of colonization and infections [57]. *S. aureus* surface proteins usually recognize and adhere to several components of the extracellular matrix (ECM) such as fibronectin, fibrin, and collagen [57].

Microbial surface components recognizing adhesive matrix molecules (MSCRAMMs) are *S. aureus* surface proteins that are anchored on the cell wall PGN. Most MSCRAMMs contain a carboxyl-terminal sorting signal containing LPXTG motif that is cleaved by *S. aureus* sortase enzymes before being covalently anchored to the cell wall PGN [58]. Notable members of this group include fibronectin binding protein A and B (FnBPA and FnBPB), collagen adhesin (Cna), staphylococcal protein A (SpA), and clumping factor A and B (ClfA and ClfB) [29].

ClfA binds to soluble fibrinogen and has been shown to inhibit complement-mediated phagocytosis [29,30]. In *S. aureus* induced septic arthritis and sepsis, ClfA is an important virulence factor that promotes the pathogenesis of the diseases and can be a target for generation of vaccine against *S. aureus* infections. Passive immunization of mice with rat and rabbit anti-ClfA antibodies gave protection against *S. aureus* induced septic arthritis and sepsis [31].

SpA not only evades the innate immunity by inhibiting opsonophagocytosis, but also has the ability to alter the response of the adoptive immunity by binding to both Fc region of IgG and Fab regions of the B-cell receptor, thereby inducing apoptosis by functioning as a B-cell superantigen [32]. SpA can also diminish the proinflammatory signaling of TNF-α by binding to its receptor, tumor necrosis factor receptor 1 (TNFR1) [59].

FnBPs are important in helping the bacteria to adhere to and invade cells of the host and together with ClfB and SpA play a role, although not yet fully understood, in forming *S. aureus* biofilms [29]. FnBPs are also involved in the pathogenesis of *S. aureus* induced sepsis [34].

Cna binds to collagen and helps to mediate the binding of *S. aureus* to cartilage [35] and was shown by Patti et al. to be involved in the pathogenesis of *S. aureus* septic arthritis [36]. Mice injected with *S. aureus* lacking Cna developed significantly less frequent signs of clinical arthritis (27%) compared with mice infected with wild type *S. aureus* (70%) [36].

*S. aureus* also secretes other proteins that, although not covalently attached to the cell wall PGN, are still surface-associated proteins and act as adhesins—they are commonly known as secreted expanded repertoire adhesive molecules (SERAMs). Coagulase (Coa), von Willebrand factor binding protein (vWbp), extracellular fibrinogen binding protein and extracellular adherence protein (Eap) are several examples of SERAMs [60]. The ability of *S. aureus* to clot human blood is mediated by the direct binding of Coa and vWbp with the hosts’ prothrombin. The resulting staphylothrombin complex will eventually convert fibrinogen to fibrin, thus forming fibrin clots [37]. Additionally, as its name suggests, vWbp acts as a bridge between the bacterial cell wall and von Willebrand factor (vWf) therefore enabling adhesion of *S. aureus* to vascular tissues [61,62]. Intriguingly, vWbp but not Coa expressed by *S. aureus* facilitates the initiation of septic arthritis and such an effect might be mediated through its interaction with a host factor (von Willebrand factor), strongly suggesting bacterial adherence to blood vessels is more important than fibrin clotting function of coagulases in induction of septic arthritis [38]. Eap plays multiple roles in *S. aureus* infections such as acting as an adhesin, inhibiting wound healing and being involved in biofilm formation [39,40].

### 1.3. Secreted Proteins

*S. aureus* superantigen like proteins (SSLs) are another group of proteins secreted by *S. aureus* that have similar structures as superantigens but lack superantigenic activities. Several SSLs have been identified that are able to interfere with the innate immune response [40,63]. Of these, SSL3 can bind to TLR2 and inhibit the production of TNF-α by macrophages stimulated by heat-killed *S. aureus* or PGN [40,63].

*S. aureus* also secretes numerous other proteins such as the chemotaxis inhibitory protein of *S. aureus*, staphylococcal complement inhibitor, and formyl peptide receptor-like-1 inhibitory protein that aid the bacteria to evade opsonization and phagocytosis [2].

Enzymes secreted by *S. aureus* include catalase, proteases, hyaluronidase, lipases, nucleases, and staphylokinase. Apart from exploiting host tissues and converting them into nutrients for the bacteria, *S. aureus* enzymes also facilitate invasion and evasion of the immune system [2]. Hyaluronidase breaks down hyaluronic acid that holds the cells of the body together, thus facilitating the invasion of *S. aureus* further into tissues [64]. Staphylokinase, which mediates the digestion of fibrin clots via activation of plasminogen to plasmin, has been shown to promote the establishment of *S. aureus* skin infections, but at the same time decrease the severity of the disease [41]. Intriguingly, fibrinolysis activated by staphylokinase prevents biofilm formation and promotes detachment of biofilms [65].

### 1.4. Toxins

The secretion of toxins is another virulence weapon of *S. aureus* that bacteria use to manipulate and gain the upper hand against the immune system. *S. aureus* secretes large amounts of toxins with several different virulence factors. Several toxins secreted by *S. aureus* have superantigenic properties, i.e., the ability to cause non-specific activation of T-cells, leading to massive polyclonal T cell activation followed by a vast release of cytokines with subsequent fever, shock, and multiple organ failure [66,67]. It was long assumed that superantigens bind only to the T cell receptor (TCR) on the T-cells and major histocompatibility complex (MHC) class II molecules on antigen presenting cells (APCs) [68]. However, it has since emerged that superantigens can also bind to CD28, thus forming a more stable complex than previously thought [69].

Toxic shock syndrome toxin 1 (TSST-1), staphylococcal enterotoxins (SEs) (AE, G-1, R and T), and staphylococcal enterotoxin like toxins (SEls) (J-Q, S, U, V, and X) are all superantigenic toxins produced by *S. aureus* [42]. Of the SEs, SEB and SEC are known to cause non-menstrual toxic shock syndrome (TSS) [45]. Furthermore, SEs have long been known to cause food poisoning whereas SEls were thought not to have emetic properties. However, recent studies have found that some newly discovered SEls (I-Q) have emetic properties and may also play some role in staphylococcal food poisoning [46]. TSST-1 accounts for almost half of all non-menstrual TSS in the general population and almost all cases of menstruation associated TSS [43]. In addition, clonal expansion of CD4+ Vβ11+ T cells induced by *S. aureus* producing TSST-1 toxin has been shown to be involved in the pathogenesis of *S. aureus* septic arthritis [44].

Another set of toxins secreted by *S. aureus* includes the hemolysins (also known as alpha (α), beta (β), and gamma (γ) toxins), cytolytic peptides (phenol soluble modulins (PSMs)) and bi-component leukocidins (including Panton-Valentine leukocidin (PVL)). All are characterized by their ability to cause cell lysis by forming pores in the cell membrane [47].

The first to be discovered and most studied of the hemolysins is the α-toxin, with an ability to form pores and lyse a broad range of cell types such as peripheral blood monocytes, platelets, and keratinocytes, and cells of the endothelium [70]. In addition to α toxins, γ toxins produced by *S. aureus* are also a critical virulence factor in *S. aureus* induced septic arthritis since mice injected with a mutant lacking both α and γ toxins showed significantly less frequent and less severe arthritis compared to wild-type strains producing both α and γ toxins [48].

As mentioned, PVL has adhesion properties, damages leukocytes and has also been implicated in the pathogenesis of necrotizing pneumonia [49].

The PSMs consist of small peptides and similar to the hemolysins and leukocidins have pore-forming properties. The PSMs can target several cell types such as erythrocytes and leukocytes and have been shown to induce inflammation [50]. Members of this family include the PSM-mec, PSMα 1-4, PSMβ 1-2, and PSMγ [50]. Apart from the role in biofilm structuring and dispersal [50,51,52], PSMs facilitate invasion and killing of osteoblasts thereby aggravating *S. aureus*-induced osteomyelitis [53]. PSMs are largely produced by community-associated methicillin-resistant *S. aureus* (CA-MRSA). Furthermore, they are involved in biofilm formation and are known to cause aggressive *S. aureus* infections [52].

### 1.5. Bacterial DNA

Bacterial but not mammalian DNA can induce inflammatory response. The proinflammatory properties of bacterial DNA are largely dependent on bacterial CpG motifs that are unmethylated cytosine–phosphate–guanine (CpG) dinucleotides. The CpG motifs are predominantly prevalent only in bacterial DNA, but not in mammalian DNA [71]. *S. aureus* DNA can induce the production of proinflammatory cytokines such as TNF-α, IL-6, and interferon-gamma (IFN-γ) via TLR9 [54]. Indeed, the injection of *S. aureus* DNA in mice led to rapid activation of macrophages followed by massive release of TNF-α that triggered lethal shock in mice [56]. Furthermore, previous results showed that *S. aureus* DNA containing CpG motifs induced arthritis [55] and meningitis through NF-κB [72]. However, our recent data suggest that DNA from antibiotic-killed *S. aureus* plays a minor role in mediating arthritis [73].

## 2. The Immune Response during *S. aureus* Infections

*S. aureus*, through its vast virulence factors, seeks multiple ways to colonize and establish infections in humans. However, upon intrusion, this pathogenic bacterium highly alerts the host’s immune system. Consequently, a battle between the host and the pathogen starts. The role of different immune cells in *S. aureus* septic arthritis and sepsis is summarized in Table 2.

### 2.1. Innate Immunity

The host’s innate immune system immediately executes a series of protective measures against intruding pathogens, such as *S. aureus*, and this serves as the first line of defense [83]. This action is initially implemented through recognition via pattern recognition receptors (PRRs) that distinctively sense pathogenic components and promptly trigger the activation of innate immune cells [83]. Among these immune cells are the phagocytes.

### 2.2. Neutrophils

Neutrophils are the most abundant type of white blood cells (WBCs) in the body, constituting around 50–70% of all WBCs and play a very important role in the innate immunity. During *S. aureus* infections, neutrophils are quickly recruited from the blood and migrate to the infection site via a process known as chemotaxis [84]. Neutrophils have several PRRs, such as Toll like receptors (TLRs) that can recognize different conserved molecules from microbes, so called pathogen-associated molecular patterns (PAMPs). At the infection site, PRRs on the neutrophils will recognize PAMPs from bacteria, which is subsequently internalized. Around the internalized bacteria, a cellular compartment known as phagosome will be formed, which fuses with lysosomes to form phagolysosomes. The rapid release of reactive oxygen species through oxidative burst, antibacterial peptides that have microbicidal effects, proteinases that degrade bacterial components, and proteins that sequester essential bacterial nutrients are some of the mechanisms employed by the neutrophils in the phagolysosomes to neutralize the internalized bacteria [85].

Neutrophils also possess the ability to kill bacteria extracellularly by releasing its contents and DNA, known as neutrophil extracellular traps (NETs). This process involves forming a web-like structure interconnected with histones and containing antimicrobial agents such as defensins and myeloperoxidase that trap the bacteria and eliminate it [86]. Neutrophils are killing machines that do an excellent job phagocytizing bacteria and thus have a short life span (1–2 days) as a regulatory precaution to avoid tissue damage. Neutrophils are absolutely essential in protecting the host against live *S. aureus* infections, as clearly exhibited by the significantly higher mortality and more severe arthritis caused by *S. aureus* in neutrophil-depleted mice compared to wild-type controls [74]. In addition, in the murine *S. aureus* skin infection model, Mölne et al. demonstrated that neutrophil depletion worsens the disease severity with increased bacterial burden in the skin tissue [87]. On the other hand, the depletion of neutrophils did not have any impact in arthritis caused either by antibiotic-killed *S. aureus* or *S. aureus* Lpp [14,73].

### 2.3. Macrophages

Macrophages are outstanding phagocytes that not only eliminate *S. aureus* but also function as APCs and are involved in activating the adaptive immunity in case of serious breaches. Activated macrophages are also potent secretors of the proinflammatory cytokine TNF-α, whose role in *S. aureus* infections will be described later.

Two distinct subtypes of macrophages have been described with opposing activities: M1 macrophages and M2 macrophages. M1 macrophages, also known as “classically activated” macrophages, are proinflammatory. The enzyme nitric oxide synthase (iNOS) is expressed by M1 macrophages and helps convert arginine into nitric oxide (NO), which inhibits proliferation of infected cells [88,89]. Microbial products, such as lipopolysaccharides (LPS) or the proinflammatory cytokine IFN-γ, stimulate the M1 macrophages phenotype that will result in a Th1 immune response [90]. This will lead to the production of more proinflammatory cytokines such as IL-12, TNF-α, and IFN-γ in a positive feedback loop, thus maintaining the M1 macrophage phenotype. M2 macrophages, or “alternatively activated macrophages”, are anti-inflammatory and give rise to a Th2 immune response and thus promote cell proliferation and wound repair. The anti-inflammatory cytokine IL-4 promotes the differentiation of macrophages into M2 macrophages and stimulates the production of IL-10, which further enhances the phenotype of M2 macrophages [91].

Macrophages have specific names depending on the tissue on which they reside. For example, Kupffer cells are macrophages that are found in the liver, whereas microglia, adipose tissue macrophages, and osteoclasts are found in the central nervous system, adipose tissue, and bones, respectively [92]. Osteoclasts and osteoblasts play an important role in maintaining bone homeostasis by degrading and synthesizing bones, respectively [93]. IL-15, a proinflammatory pleiotropic cytokine, plays a potent role in early osteoclast differentiation [94]. Additionally, the receptor activator of nuclear factor kappa-B-ligand (RANKL)-dependent osteoclastogenesis is known to be impaired in the mice lacking the IL-15 receptor [95]. In *S. aureus* septic arthritis, mice lacking IL-15 were found to have a reduced number of osteoclasts in their joints, which also coincided with reduced arthritis severity and less joint destruction compared to wild-type mice [96]. Activation of osteoclasts requires the RANKL, a member of the tumor necrosis factor (TNF) superfamily that is found on the surface of osteoblasts, to bind to receptor activator of nuclear factor kappa-B (RANK) on the surface of osteoclasts. RANKL has been implicated in *S. aureus* infections, and its inhibition reduces bone loss in *S. aureus* septic arthritis [97].

Macrophages have been shown to play dual roles in *S. aureus* infections. On the one hand, Verdrengh et al. showed that macrophages are involved in aggravating *S. aureus* arthritis and the deficiency of macrophages attenuated the disease [75]. On the other hand, the ability of the host to clear invading bacteria in the kidneys is impeded, thus leading to higher mortality [75]. Further studies also showed that macrophages are involved in arthritis triggered by bacterial DNA containing CpG motifs [98]. We recently demonstrated that purified *S. aureus* Lpp rapidly initiates the recruitment of monocytes/macrophages and neutrophils upon local knee injection [14]. In the model of local knee arthritis induced by purified *S. aureus* Lpp, depletion of monocytes/macrophages resulted in diminished bone destruction [14]. This demonstrates that monocytes/macrophages are the key cell types in the development of local knee arthritis induced by purified Lpp. Similarly, double depletion of both neutrophils and monocytes abrogated the arthritis induced by antibiotic-killed *S. aureus* [73].

### 2.4. Natural Killer (NK) Cells

NK cells are a type of white blood cells that play an important role in the innate immune system. NK cells respond to and eliminate virus-infected and tumor cells and do not require antibodies or MHC to respond to these cells. NK cells play a protective role during *S. aureus* infections [76,99]. The depletion of NK cells in mice before inoculation with a TSST-1 secreting strain of *S. aureus* is associated with higher susceptibility to develop *S. aureus* septic arthritis as compared to control mice with intact NK cells [76]. Further studies have also shown that mice depleted of NK cells are significantly more susceptible to pulmonary *S. aureus* infections compared to wild-type mice [99], underscoring the protective role of NK cells against *S. aureus* infections.

### 2.5. The Complement System

The complement system serves as the first line of defense and is a crucial part of the innate immune system. It is made up of several plasma proteins and can be activated through three different pathways: the classical, the alternative and the lectin pathway. Whenever bacteria are successful in breaching the physical barriers, the complement system, regardless of the activation pathway, will recognize this and form enzyme complexes known as C3 convertases whose task is to cleave the complement component 3 (C3) into two different proteins. The C3a, a proinflammatory anaphylatoxin, helps with the recruitment of the phagocytes to the infection site, whereas the C3b opsonizes the invading *S. aureus,* thus making it easier to be phagocytosed [100].

Apart from opsonizing the bacteria, the complement system can also form a lytic complex known as the membrane attack complex (MAC) on the surface of invading bacterial cells that will lead to the lysis and eventual death of the microbe. However, the MAC recognizes only Gram-negative bacteria and thus *S. aureus* is spared from the potent killing ability of the MAC mechanism [101].

The complement system is imperative to the host defense during *S. aureus* infection as its deficiency renders the host defenseless and significantly increases the susceptibility to *S. aureus* infections [102]. Recent data from our lab show that mice lacking the complement component 3 (C3-/-) are highly susceptible to *S. aureus* septic arthritis. Kidney abscess formation and bacterial burden in the kidneys are also negatively affected in the C3-/- mice compared to the wild type controls with functioning complement system [103]. The results underscore the importance of the complement system in fending off *S. aureus* infections.

### 2.6. Adaptive Immunity

#### 2.6.1. T-Cells

T-cells or T-lymphocytes are an integral part of adaptive immunity. They originate in the bone marrow but mature in the thymus, hence the name T-cells. T-cells are recognized from other lymphocytes due to their unique TCR displayed on the cell surface.

T-helper (T_h_) cells (CD4^+^ T-cells) express CD4 glycoprotein on their surface, recognize antigens presented by MHC class II and secrete cytokines that are necessary for both the cell-mediated and humoral immune response [104]. Cytotoxic T-cells (CD8^+^ T-cells) express CD8 glycoproteins, recognize antigens presented by MHC class I and eliminate virus-infected and tumor cells [104]. Regulatory T-cells (Tregs) play an important role in maintaining balance by preventing the immune response to self-antigens and suppressing excessive immune response that can cause autoimmune diseases [105]. T-helper 17 (Th17) cells are a unique CD4^+^ T-cell subset characterized by the production of IL-17 that is a highly inflammatory cytokine playing an important role in the pathogenesis of several autoimmune diseases [106].

CD4^+^ T-cells differentiate into two major subgroups: Th1 and Th2 cells. Th1 cells mainly secrete the cytokines IFN-γ and IL-2, respond to intracellular microbes and stimulate phagocyte mediated uptake and elimination of microbes [107,108]. Th2 cells usually respond to extracellular pathogens such as gastrointestinal parasites, secrete mainly IL-4 and IL-5 cytokines and promote eosinophil activation and phagocyte-independent immune response [107]. Cytotoxic T-lymphocyte-associated protein 4 (CTLA4), a naturally occurring protein receptor expressed on the surface of the T-cells, has the ability to inhibit the activation of the T-cell by competitively binding to CD80/86. CTLA4-Ig, a biologic that inhibits the full activation of T-cells, downregulates the Th2 response, and has little effect on Th1 response [109]. Septic arthritis mice pretreated with CTLA4-Ig exhibited more severe joint inflammation but lower levels of IL-4 compared to the control mice [110]. Although both CD4^+^ and CD8^+^ T cells are found in the inflamed synovium, CD4^+^ T cells make up the overwhelming part. Furthermore, depletion of CD4^+^ cells significantly ameliorates the course of septic arthritis in mice, whereas depletion of CD8^+^ T cells does not alter the course of arthritis compared to control mice [78]. Thus, it appears that CD4^+^ T cells are pathogenic during *S. aureus* septic arthritis due to their ability to produce proinflammatory cytokines such as TNF-α and IFN-γ via activated macrophages [78]. Recent results also found that CD4^+^ T cells promote the pathogenesis of *S. aureus* pneumonia and *Pseudomonas aeruginosa* septic arthritis [111,112].

The depletion of Tregs by anti-CD25 monoclonal antibodies aggravated the severity of septic arthritis without impact on bacterial clearance in mice [79], suggesting the protective role of Tregs in development of septic arthritis. The role of Th17 cells in *S. aureus* septic arthritis and sepsis is still unclear. However, IL-17 produced by Th17 cells is crucial for host defense against *S. aureus* skin infections [113] and septic arthritis [114].

#### 2.6.2. Natural Killer T (NKT) Cells

NKT cells are a unique subset of T cells that can have features of both T cells and NK cells. While other subsets of T cells recognize protein antigens, NKT cells are unique in that they recognize lipids and glycolipids and make up a tiny percentage of blood T cells. Studies from *S. aureus* triggered sepsis indicate that NKT cells do not play any significant role in the course of the disease [115].

#### 2.6.3. B-Cells

Unlike T-cells, B-cells do not seem to be the driving force behind the pathogenesis of *S. aureus* infections. Studies from murine *S. aureus* septic arthritis model showed that B-cell deficient mice do not differ from the wild-type controls with regards to arthritis and clearance of the bacteria, but tended to have higher mortality [80].

#### 2.6.4. Other Cell Types

Innate lymphoid cells (ILCs) are enriched at barrier surfaces of hosts and play critical roles in maintaining tissue homeostasis and immune defense. ILCs are divided to three groups of cells with distinct immunological functions [116]. Recently, activation of type 2 ILCs (ILC2s) were shown to be protective against *S. aureus* sepsis in mice by promoting eosinophilia, enhancement of type 2 immunity, and consequent balance of dysregulated septic inflammatory responses [77].

Basophils are the rarest granulocyte (<1% of peripheral blood leukocytes). A very recent study demonstrated that basophil-deficient mice exhibited reduced bacterial clearance and increased mortality in a cecal ligation and puncture model of sepsis and such effect was due to reduced basophil-derived TNF production [82]. So far, the role of basophils in *S. aureus* septic arthritis and sepsis is still largely unknown.

Thrombocytes are the major effector cell in hemostasis. Interestingly, they also contribute to the protection against *S. aureus* infections through direct bacterial killing effect and by enhancing phagocytic capacity of macrophages [117]. Not surprisingly, thrombocyte depletion gave rise to reduced bacterial clearance and increased mortality in *S. aureus* systemic infections [81].

#### 2.6.5. Cytokines

Several cytokines are secreted by cells of both the innate and the adaptive immunity and have different roles in *S. aureus* infections. Some of them will briefly be discussed below and are listed in Table 3.

TNF-α, one of the most studied cytokines due to its role in inflammation and many diseases, is involved in the acute-phase reaction and is mainly secreted by activated macrophages and CD4+ cells, neutrophils, mast cells, and NK cells [127].

TNF-α has a contrasting role during *S. aureus* infections. In patients with *S. aureus* arthritis, the levels of TNF-α have been shown to be highly elevated in the synovial fluid. Furthermore, it has been suggested that the levels of the cytokine could function as a predictor in determining the prognosis of the disease, with higher levels associated with worse prognosis [128]. Animal studies have shown that TNF/lymphotoxin (LT)-α double knockout mice have significantly less severe *S. aureus* arthritis compared to the wild-type mice [118]. Indeed, we could also show that TNFR1 knockout mice exhibited less arthritis compared to wild-type mice in antibiotic-killed *S. aureus* induced arthritis [73]. Anti-TNF treatment was also able to abrogate arthritis induced by antibiotic-killed *S. aureus* [73]. Additionally, in the *S. aureus* skin infection model, mice pretreated with anti-TNF agent exhibited smaller lesion (abscess) sizes compared to the control PBS-treated mice [129]. On the other hand, the lack of TNF-α was associated with impaired ability of the host to successfully clear invading *S. aureus* in the kidneys [110,118], thus leading to increased mortality [118].

IL-1 cytokine family is a group of eleven cytokines that play an important role in the inflammatory response. Of these, most is known regarding IL-1α IL-1β, and IL-1 receptor antagonist (IL-1Ra). IL-1α plays a central role in the induction of fever, sepsis, and inflammation and is produced by activated macrophages, neutrophils, and endothelial and epithelial cells. IL-1β is predominantly produced by activated macrophages as a proprotein and is cleaved by caspase 1 into its active mature form [130]. It plays an important role in pain, inflammation, and cartilage degradation in several inflammatory diseases [131]. In *S. aureus* systemic infections, IL-1R signaling is also essential to the host protection against the bacteria as shown by Hultgren et al. IL-1R^−^/^−^ mice inoculated with *S. aureus* developed significantly higher *S. aureus* septic arthritis and sepsis compared to wild-type IL-1R^+^/^+^ mice [119].

IFN-γ, a potent proinflammatory cytokine, is mainly produced by NK cells and T-cells. Apart from inhibiting viral and even bacterial infections, IFN-**γ** activates and stimulates the macrophages to better phagocytose intracellular invaders. Different roles of IFN-**γ** in *S. aureus* triggered sepsis and septic arthritis have been described. Mice deficient of the IFN-γ receptor develop significantly more severe and frequent arthritis [132]. The mortality levels due to sepsis are also significantly increased during the early stages of the infection in the mice lacking IFN-γ receptor, whereas in later stages the reverse is true with higher mortality levels in the wild-type mice [132]. Likewise, in vivo administration of IFN-γ before and after inoculation of *S. aureus* improved the survival of the mice while at the same time increased the severity and frequency of arthritis [125]. The positive effects on mortality due to in vivo administration of IFN-**γ** correlated with improved phagocytosis and better clearance of the bacteria in both, the liver and the kidneys. On the other hand, treatment of the mice with anti-IFN-γ monoclonal antibodies attenuated the severity and frequency of arthritis due to lower levels of serum TNF-α, IL-6, and IL-1β [125].

IL-4 is an anti-inflammatory cytokine that has a role in differentiation of naïve T-cells into Th2 cells and the differentiation of B-cells into plasma cells. IL-4 promotes the cytotoxic activity of CD8+ cells, decreases the production of IFN-γ by T-cells and NK cells, and affects monocytes/macrophages by reducing their production of proinflammatory cytokines like IL-1, IL-6, and TNF-α [133]. IL-4 inhibits the intracellular killing of *S. aureus* in infected macrophages, without affecting phagocytosis and provides therefore a favorable milieu for survival of staphylococci [122]. In *S. aureus* infections, the dual role of IL-4 has been described depending on the genetic background of the host. In inbred C57BL/6 mice, IL-4 was shown to be a driving force of septic arthritis and sepsis by significantly impairing the capability of the host to clear the bacteria [122]. Enhanced staphylococcal clearance from joints and kidneys, reduced mortality, and decreased frequency of arthritis was observed in IL-4 deficient C57BL/6 mice. However, in another inbred strain, 129SV mice, the opposite was true, i.e., IL-4 protected the mice from *S. aureus* induced sepsis [123]. IL-4 deficient 129SV mice had a thousand times higher bacterial growth in their kidneys, significantly elevated mortality, and delayed development of septic arthritis. A differential pattern of host responsiveness was seen between these mouse strains and explanation for the discrepant outcome could lie in the variation in circulating IL-4 levels—serum IL-4 was not detectable in C57BL/6 mice whereas increased IL-4 production was observed in 129SV mice in response to *S. aureus* infection [123].

Although IL-6 has been shown to have some anti-inflammatory features, it is usually regarded as a proinflammatory cytokine [134]. Macrophages and T cells mainly produce IL-6 during infections or trauma. In *S. aureus* infections, IL-6 is usually elevated together with other proinflammatory cytokines such as IL-1β and TNF-α [128]. Synovial IL-6 together with synovial lactate and synovial fluid white blood cells count have been touted as good parameters for diagnosing septic arthritis [120].

IL-10 is an anti-inflammatory cytokine produced mainly by monocytes and to a smaller extent the lymphocytes. IL-10 promotes Th2 response while downregulating Th1 cytokine secretion by macrophages and monocytes. IL-10 plays a crucial role protecting the host against *S. aureus* septic arthritis by promoting bacterial clearance [124].

IL-12 is primarily produced by monocytes, macrophages, and dendritic cells. Apart from stimulating the differentiation of naive T-cells to Th1 cells, IL-12 is also involved in the production of IFN-γ and TNF-α via T-cells and NK-cells. In *S. aureus* infections, IL-12 is crucial for the survival of the host and deficiency of IL-12 is associated with significant accumulation of *S. aureus* in many organs leading ultimately to the demise of the host [121].

IL-17A is a proinflammatory cytokine produced by activated Th17 subset of T-cells. IL-17A plays a significant role in host defense against local *S. aureus* infections due to its ability to induce chemokines that attract and recruit neutrophils [135]. Thus, in local *S. aureus* infection, IL-17A-/- mice developed more synovitis and erosions and more weight loss compared to the wild-type mice [114]. On the other hand, IL-17A-/- mice did not differ from wild-type mice regarding the severity and the frequency of arthritis induced by antibiotic-killed *S. aureus* [73].

## 3. Septic Arthritis and Sepsis

Septic arthritis is a rapidly progressing and devastating joint disease caused by pathogen infection. *S. aureus* accounts for about 70% of the septic arthritis cases and has been shown to cause more severe infection than other microbes [136,137]. Prevalence of septic arthritis is around 6 cases per 100,000 in the general population and much higher in rheumatoid arthritis (RA) patients approaching about 70 cases per 100,000 [137]. The mortality rate is around 10–15% in non-RA patients with monoarthritis, i.e., arthritis in a single joint. Polyarthritis, on the other hand, is associated with much worse prognosis, with the mortality rate ranging up to 30–50% [137,138]. Risk factors for septic arthritis include: increasing age, pre-existing joint diseases (especially RA), intravenous drug abuse, prosthetic joints, and diabetes mellitus [137,138]. Treatment of septic arthritis consists primarily of antibiotics and joint aspiration to flush out the intra-articular pus containing both bacteria and infiltrating immune cells [139,140]. One of the devastating aspects of septic arthritis is that despite optimal antibiotic treatment, almost half of the patients will develop irreversible joint destruction [138]. Definitive diagnosis of septic arthritis requires the isolation of the microbe from the synovial fluid, although due to the fast progressing nature of the disease, physicians do not and should not wait for culture results before initiating treatment with broad spectrum antibiotics [139].

Hematogenous spread of *S. aureus* to the synovial membrane of joints is the most commonly reported route of acquiring septic arthritis, although the bacteria can also be introduced directly into the joints by trauma (e.g., needle accident) or spread from neighboring inflamed tissues [140]. In a retrospective study, more than 70% of septic arthritis cases were shown to be caused by hematogenous spreading [141]. The probability to develop septic arthritis following bacteremia with the optimal arthritogenic dose of *S. aureus* in our animal model can reach up to 80–90%. In patients, the frequency of the bone and joint infections after *S. aureus* bacteremia varies from 12 to 17% in different studies [142,143]. Once inside, the bacteria will employ different virulence factors to attach to the host tissue and proliferate while the host immune system will respond to the invading bacteria. It has been shown that the destruction of joints in *S. aureus* septic arthritis is not only caused by the invading microbes, but also by cells and molecules of the immune system, involving both the innate and adaptive immunity [140].

Sepsis is defined as the systemic inflammatory response due to an infection and is usually caused by bacteria such as *S. aureus*, *Pseudomonas aeruginosa,* and *Escherichia coli* [144]. *S. aureus* bacteremia is associated with higher mortalities than bacteremia caused by most other microbes and can develop to sepsis and severe sepsis [144]. Despite advances made in critical care and treatment, sepsis remains one of the foremost causes of death in critically ill patients. Mortality in sepsis is around 10-20% and increases significantly up to 80% if a septic shock develops [145,146].

The pathogenesis of *S. aureus* sepsis is multifactorial and is mediated by components of the bacteria and the exaggerated immune response mounted by the host. Bacterial superantigens can cause non-specific activation of T-cells leading to massive polyclonal T cell activation and resulting in a vast release of proinflammatory cytokines such as TNF-α and IL-1β [66,67]. PGN and LTA, cell wall components of *S. aureus*, can also interact with CD14 molecules through TLR2 and stimulate the release of proinflammatory cytokines (TNF-α and IL-6) and chemokines (IL-8) further potentiating the systemic inflammation in sepsis [54,147]. This is followed by a massive release of anti-inflammatory cytokines in response to the inflammation whereby the immune regulation is rendered inactive, leading to a state of immunosuppression [148]. Without a proper functioning immune system, the bacteria have free reign to proliferate and spread to different organs. Coagulation disorder, characterized by an excessive coagulation, is another attribute of sepsis. The coagulation cascade can be activated through proinflammatory cytokines such as TNF-α, IL-1β, and IL-6 leading to disseminated intravascular coagulation (DIC) [147,149]. DIC is soon followed by thrombocytopenia, i.e., the lack of platelets in the blood resulting in massive bleeding from several sites and leading to organ failure [149]. Given together, the pathogenesis of sepsis includes systemic inflammation, loss of immune regulation, and excessive coagulation that altogether will lead to multiple organ failure, shock, and finally the demise of the host.

## 4. Bacteria and Host Interplay Determines the Disease Outcome

### 4.1. Joint Affinity of S. aureus is the Key Mechanism of Septic Arthritis

The majority of septic arthritis has the hematogenous origin. The course of the disease can be divided into two stages—early and late. During the early stage, *S. aureus* needs to adapt to the host environment, to survive the bactericidal components and phagocyte attacks in the blood, to disseminate to synovial tissue, and finally to reach the joint cavity. In the joint cavity (late stage), *S. aureus* proliferates and releases a vast arsenal of components that arouse a host response and cause joint damage. Bacterial joint invasion is the key step of disease mechanism of septic arthritis, since the initial cause of disease is the invading *S. aureus* in affected joints. The defect of innate immunity on the host side increases the susceptibility to *S. aureus* septic arthritis, which is correlated to impaired bacterial clearance. Several bacterial factors are known to determine the joint-invading process of *S. aureus*. Actually, so far all of them are bacteria surface proteins including clumping factors, protein A, and Cna. Unlike *S. aureus*, the related coagulase-negative staphylococci (CoNS) are hardly found in native septic arthritis, suggesting coagulases might play a major role in the disease pathogenesis. Indeed, our very recent data demonstrate that the depletion of vWbp but not Coa almost fully abolished the capability of *S. aureus* to invade joint cavity but no other organs. Such effect is fully dependent on the vWf expression in the host and strongly suggests that the interaction between vWbp and vWf determines joint-specific invasiveness of *S. aureus* [38]. It seems that expression of *S. aureus* surface proteins that “stick to” blood vessel wall or joint/cartilage is the key mechanism of disease initiation.

### 4.2. The Impact of Bacteria and Host Interplay on Biofilm Formation in Septic Arthritis

Biofilm is a type of bacterial growth characterized by formation of multicellular bacterial communities, held together by the extracellular matrix. Environmental bacteria often form biofilms and 65–80% of infections including osteomyelitis and septic arthritis according to Centres for Disease Control and Prevention (CDC) and National Institutes of Health (NIH) are also characterized by formation of bacterial biofilms [150]. The biofilm-like structure was frequently observed in the cases of joint infections [151]. In vitro, *S. aureus* aggregation was promoted and extremely thick biofilm was formed in the presence of synovial fluids [152]. Biofilm formation renders the bacteria resistant to treatment with antibiotics, as a diffusion of chemical compounds across the matrix is severely restricted. Furthermore, host defense, such as neutrophils, are unable to phagocytize bacteria in a biofilm [153]. Surface proteins of *S. aureus* are likely involved in promoting cell–cell adhesion and biofilm formation [150]. On the other hand, PSMs secreted by *S. aureus* in a quorum-sensing controlled fashion, structure biofilms, and cause biofilm detachment and bacterial dissemination [51]. In infection caused by the presence of foreign implants, the first stage of biofilm formation is the coating of the implant surface by host-derived proteins, such as fibrinogen, fibronectin, and collagen. *S. aureus* can adhere to these proteins via so called MSCRAMMs [154]. After bacterial attachment, material secreted by the microorganisms, such as DNA and polysaccharide, imbed the bacteria and protect them from environmental stress. Intriguingly, bacteria also utilize the host proteins, e.g., fibrin to build up the biofilm architectures [65,155]. With time, biofilm architecture changes, a process termed biofilm maturation [156]. Importantly, plasmin, the key fibrinolysis molecule abundant in the circulation, can digest most of host-derived adhesions and extracellular matrix proteins. We have recently shown that fibrinolysis activated by both bacterial and host plasminogen activators prevent biofilm formation and promote detachment of biofilms [65,157].

### 4.3. The Role of Microbiome in Septic Arthritis and Sepsis

Every human body is colonized by huge number of microorganisms after birth. Microbiome, a diverse community of microorganisms and the environment that they occupy, has been shown to be associated with human diseases, such as metabolic disorders, respiratory diseases, autoimmune diseases, and psychological disorders [158]. Not surprisingly, microbiome also plays a potent role in infectious diseases, as microbiota indirectly mediate colonization resistance by stimulating host mucosal immune defenses to prevent invasion of pathogens and subsequent infection. A very good example is *Clostridium difficile* (*C. difficile*) infection that is likely caused by disruption of gut microbiota and increased community susceptibility to the vegetative growth of *C. difficile* spores by antibiotic therapy [159]. At the same time, the changes of microbial community structure alter host responses to induce low levels of inflammation, which leads to greater risk for infections [160]. It is known that vaginal microbiota with low diversity (mostly *Lactobacillus* species) is associated with a lower risk of bacterial vaginosis and decreased risk of HIV-1 infection compared to high-diversity vaginal microbiota [161]. A very recent study demonstrated that mice fed with western-type diet lost *Bacteroidetes* in their gut microbiota and become susceptible to lethal postoperative infection associated with multiple drug resistant organisms present among the gut microbiota after exposure to antibiotics and an aseptic surgical procedure [162]. This study clearly demonstrated the dysbiosis as a potential risk factor for sepsis. Indeed, in elderly patients, dysbiosis was shown to be strongly associated with increased incidence of a subsequent hospitalization for sepsis [163]. Additionally, dysbiosis in patients who undergo allogeneic bone marrow transplantation was also associated with increased risk of bloodstream infection and sepsis [164].

So far, it is still largely unknown whether microbiome plays the role in initiation and development of septic arthritis. However, several lines of evidences suggest that the microbiome may contribute to the pathogenesis of autoimmune arthritis such as RA and spondyloarthritis. It has been shown that segmented filamentous bacteria, a single gut-residing microbe, is able to drive autoimmune arthritis via Th17 cells [165]. In mice with collagen induced arthritis, significant dysbiosis and mucosal inflammation occurred at the early stage of the disease. Depletion of the microbiota resulted in decreased disease severity [166]. In patients, dysbiosis with specific characteristics was evidenced in both RA and spondyloarthritis [167]. Secondary osteoarthritis is one of the most common consequences of septic arthritis. Germ-free mice were shown to be more resistant to development of osteoarthritis after joint injury [168], suggesting the potent role of gut microbiota in osteoarthritis pathogenesis. Interestingly, treatment with lyophilized inactivated culture of *Bifidobacterium longum* protected cartilage structure lesions and decreased type II collagen degradation in a spontaneous model of osteoarthritis [169]. In addition, the prebiotic fiber supplementation displayed the protective effect on osteoarthritis development in the rat model of obesity [170]. As the microbiome plays a role in both autoimmune arthritis and osteoarthritis, it is likely that the microbiome has some impact on the disease pathogenesis of septic arthritis, which should be studied in the future.

### 4.4. Exaggerated Immune Response Causes Joint Damage in Septic Arthritis

As soon as bacteria reach the joint cavity, a rapid immune response is triggered. In patients, even after they have received immediate treatment, the joint damage caused by septic arthritis is often irreversible, leading to permanent joint dysfunction for up to half of the patients [171]. Recently, we demonstrated that antibiotic-killed *S. aureus* induce destructive arthritis through TNFR1 and that bacterial cell walls are the culprits [73]. Among the *S. aureus* cell wall components, Lpp are TLR2 agonists and the main immune stimulators, while LTA are of much less importance in this respect. Lipidation of Lpp is known to be crucial for virulence in murine *S. aureus* systemic infection [22]. Our recent data suggest that *S. aureus* Lpps are one of the main instigators of joint damages in septic arthritis [14].

Focal bone destruction in autoimmune arthritis is due to excess bone resorption by osteoclast activation, which is mediated by increased local expression of RANKL compared to its decoy receptor osteoprotegerin [172]. Osteoclasts not only exist inside the bone, but can also be derived from mature monocytes and macrophages when a suitable microenvironment is provided [173]. Monocytes/macrophages have been shown to mediate bone erosions in the arthritis induced by other *S. aureus* components, such as bacterial DNA [55], PGN [12], and Lpp [14], which suggests that monocytes/macrophages are the most important immune cells in determining the outcome of septic arthritis.

To limit the immune response and reduce the risk of permanent joint destruction, a combination treatment of antibiotics and immunomodulatory therapy was proposed by us [73,174]. However, our recent data suggest that there are potential dangers associated with such combination therapies as long as the problem of antibiotic resistance persists [73,110]. Interestingly, different immunomodulatory therapies have distinct clinical outcomes. Anti-TNF therapy increased the bacterial load in kidneys but had no impact on septic arthritis development. In contrast, CTLA4-Ig treatment and IL-1 inhibitor aggravated septic arthritis but with no impact on bacteria clearance [110,175].

### 4.5. Cytokine Storm and Immune Paralysis in S. aureus Sepsis

Spread of staphylococci in the body or severe local infection, lead to systemic inflammation. Activated immune cells secrete vast amounts of proinflammatory cytokines, like IL-6 and TNF-α, which is called a “cytokine storm”. These further increase the activity of the immune system, which in extreme cases can lead to organ damage, DIC, and death. At the same time, inflammation induces expression of anti-inflammatory cytokines, such as IL-10, which are responsible for regulating the immune response. Elevated levels of both pro- and anti-inflammatory cytokines in circulation reflect severity of infection and inflammation. Already in 1987, the TNF inhibitor was shown to prevent septic shock in lethal bacteremia in baboons [176]. Not surprisingly, direct intravenous injection of TNF and IL-1 synergistically induced a shock-like status in rabbits [177]. Obviously, those proinflammatory cytokines play a potent role in induction of septic shock and lethality. We also demonstrated that the combination therapy of antibiotics and anti-TNF reduces mortality of *S. aureus* sepsis in mice [174]. Importantly, anti-TNF treatment also prevented *S. aureus* enterotoxin induced shock [174]. Not only TNF, but also other host mediators are involved in sepsis pathogenesis. A good example is the high mobility group box 1 protein (HMGB-1) as a late mediator for the endotoxin induced lethality [178]. Despite the robust anti-shock effect in the animal model, the clinical trial of anti-TNF treatment was proved to be a big failure [179]. The explanation might be that clinical trials were very heterogeneous in causative pathogens that might lead to differences in the immunological response and different stages of sepsis when the treatments were introduced. Indeed, a recent study demonstrated that RA patients treated with TNF inhibitors survived better in severe infections than those treated with conventional disease-modifying antirheumatic drugs (DMARDs). This suggests that successful immunosuppression at the very early stage of disease might be beneficial [180]. Actually, our very recent data also demonstrated that pretreatment of mice with JAK kinase inhibitor (tofacitinib) significantly prolonged the survival of mice with *S. aureus* sepsis [126]. More importantly, pretreatment of tofacitinib exerted full protection for *S. aureus* enterotoxin induced shock in mice, whereas later treatment had no effect at all [126]. These clearly show that in the clinical situation, the more personalized drug choice is crucial for better outcome in management of *S. aureus* sepsis.

It is also clear that immune paralysis directly after hyperinflammation in sepsis is another problem regarding the concept of immune suppressive treatment in sepsis [181], as patients with *S. aureus* sepsis may have already passed the hyperinflammatory stage and come in the hypoinflammatory stage. Additionally, the half-time of biologics is often long, which makes the treatment in context of sepsis even more complicated.

## 5. Therapies Targeting Host and Bacteria Interaction

Currently, three main approaches are used to treat staphylococcal infections [182]. The primary goal is the removal of infecting bacteria by using antibiotics and damaged tissues and inflammatory infiltrates—therefore abscesses are drained, infected joints undergo lavage and, if necessary, a larger scale debridement is performed in soft tissue infections. Finally, a supportive treatment is needed to maintain homeostasis if organ dysfunctions develop during infection. Possibilities of disease prevention are limited to controlling the spread of multiresistant strains, isolation of patients spreading bacteria in hospital environments, and elimination of staphylococcal colonization in high-risk groups by an aggressive chemotherapy [182]. The development of vaccine, especially those focusing on the generation of opsonic antibodies, is a history of repeated failures [183].

The development of new treatments for septic arthritis has stagnated. The current treatment alternatives are exactly the same as those used 30 years ago. To limit the immune response and reduce the risk of permanent joint destruction, a combination treatment of antibiotics and immunomodulatory therapy including non-steroidal anti-inflammatory drug [184,185], corticosteroids [186], and TNF inhibitors [174] has been tested in the animal models. Indeed, in patients two double-blind randomized controlled trials showed a positive effect for the addition of corticosteroids to antibiotics in the treatment of septic arthritis [187]. However, larger randomized controlled trial with long term follow up and safety data is still missing. Therefore, we do not have enough evidence to recommend corticosteroids alongside antibiotics for the treatment of septic arthritis. Actually, there are potential dangers associated with such combination therapies as long as the problem of antibiotic resistance persists [110,175]. Dissecting the interaction between bacterial components and host factors responsible for joint inflammation and destruction is the key for the development of new therapies. Our recent data demonstrate that interaction between vWbp and vWf mediates the joint-specific invasiveness of *S. aureus* [38]. We suggest that three distinct strategies might therefore be used in the future studies to block the interaction between vWf and vWbp: (a) passive and active immunization against vWbp; (b); cleavage of vWbp by recombinant ADAMTS 13 (a disintegrin and metalloproteinase with a thrombospondin type 1 motif, member 13); and (c) specially designed peptides interfere with binding site of vWbp/vWf. 

## 6. Concluding Remarks

The interplay between bacteria and host determines the outcome of *S. aureus* septic arthritis. Innate immunity including neutrophils and complement system plays a protective role in the development of septic arthritis. The most reasonable explanation for this is that efficient innate immune killing leads to lower bacterial concentration in the blood stream, which gives less chance to bacterial entry to the joints. In contrast, the role of adaptive immunity in septic arthritis is less clear than innate immunity. On the bacterial side, *S. aureus* surface proteins mediate the joint invasiveness of bacteria via interaction with the blood vessel surface and joint components, which usually does not cause bacterial overgrowth in organs. As soon as bacteria reach the joint cavity, the bacterial components such as Lpp arouse a strong immune response with rapid recruitment of neutrophils and monocytes. The bone destruction in septic arthritis is closely related to monocyte/macrophage activation and differentiation. The future therapeutic strategies against septic arthritis might be combination therapy of antibiotics and anti-monocyte activation treatment.

In *S. aureus* sepsis, cytokine storm (highly elevated TNF, IL-1, and IL6) induced by *S. aureus* superantigens, Lpp, PGN, and other components causes hyperinflammation, DIC, multiple organ failure, and later death. The immune suppressive therapies at the very early time point might be protective. However, the timing of treatment is crucial, as late treatment may aggravate the immune paralysis and lead to uncontrolled infection and death.

## Figures and Tables

**Figure 1 pathogens-10-00158-f001:**
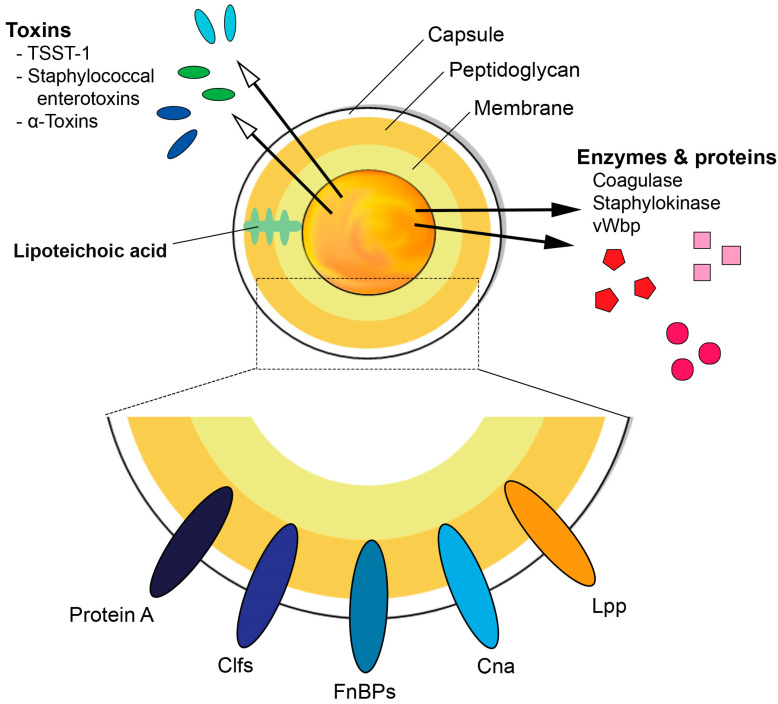
Schematic diagram illustrating the basic structure of *Staphylococcus aureus* and its ability to express various virulence factors. TSST-1 = Toxic shock syndrome toxin-1, Clfs = clumping factors, FnBPs = fibronectin binding proteins, Cna = Collagen adhesin, Lpp = lipoproteins, vWbp = von Willebrand factor-binding protein.

**Table 1 pathogens-10-00158-t001:** *S. aureus* virulence factors and their roles in infections.

	Biological Functions	Role in Infection
**Cell Wall Components:**		
**Capsular Polysaccharide**	Antiphagocytic [4]	Septic arthritis, sepsis [5]
**Peptidoglycan**	Release of TNF-α, IL-6 [12]	Arthritis [12], bone loss [13]
**Lipoproteins**	Activation of innate immunity [25]	Arthritis [14], sepsis [27]
**Surface Proteins:**		
**ClfA**	Inhibit complement-mediated phagocytosis [29,30]	Septic arthritis, sepsis [31]
**SpA**	Inhibit opsonophagocytosis, B-cell superantigen [32]	Septic arthritis, sepsis [33]
**FnBPs**	Adhesion and invasion of cells [29]	Biofilm formation [29], sepsis [34]
**Cna**	Mediates binding of *S. aureus* to cartilage [35]	Septic arthritis [36]
**Secreted Proteins:**		
**vWbp**	Promote blood-clotting [37]	Septic arthritis [38]
**Eap**	Adhesin [39,40]	Inhibit wound healing, biofilm formation [39,40]
**Staphylokinase**	Mediates digestion of fibrin clots [41]	*S. aureus* skin infections [41]
**Toxins:**		
**TSST-1**	Superantigen [42]	Toxic shock syndrome [43], septic arthritis [44]
**SE**	Superantigen [42]	Toxic shock syndrome [45], food poisoning [46]
**SEls**	Superantigen [42]	Food poisoning [46]
**α- and γ-toxins**	Cell lysis [47]	Septic arthritis [48]
**PVL**	Cell lysis [47]	Necrotizing pneumonia [49]
**PSMs**	Cell lysis [47], induce inflammation [50]	Biofilm formation [50,51,52], osteomyelitis [53]
**Bacterial DNA**	Release of TNF-α, IL-6, IFN-γ [54]	Arthritis [55], septic shock [56]

Abbreviations: ClfA = Clumping factor A; SpA = Staphylococcal protein A; FnBPs = Fibronectin binding proteins; Cna = Collagen adhesion; vWbp = von Willebrand factor binding protein; Eap = extracellular adherence protein; TSST-1 = Toxic shock syndrome toxin 1; SE = Staphylococcal enterotoxins; SEls = Staphylococcal enterotoxin like toxins; PVL = Panton-Valentine leucocidin; PSMs = Phenol soluble modulins.

**Table 2 pathogens-10-00158-t002:** The role of different immune cells in *S. aureus* septic arthritis and sepsis.

Cell Types	Septic Arthritis	Bacterial Clearance	Sepsis
**Neutrophils**	Protective [74]	Enhance [74]	Pathogenic [74]
**Monocytes/macrophages**	Pathogenic [75]	Enhance [75]	Protective [75]
**NK cells**	Protective [76]	Enhance [76]	Protective [76]
**Innate lymphoid cells**	NA	NA	Type 2 ILCs are protective [77]
**CD4+ T cells**	Pathogenic [78]	No effect [78]	Pathogenic [78]
**CD8 T cells**	NA	NA	NA
**Regulatory T cells**	Protective [79]	No effect [79]	NA
**Th17 T cells**	NA	NA	NA
**B cells**	No effect [80]	No effect [80]	Protective? [80]
**Thrombocytes**	NA	NA	Protective [81]
**Eosinophils**	NA	NA	Protective [77]
**Basophils**	NA	NA	Protective [82]

NA = not assessed; ILCs = innate lymphoid cells.

**Table 3 pathogens-10-00158-t003:** The role of cytokines in *S. aureus* septic arthritis and sepsis.

Cytokine	Cell Source	Function	Role in *S. aureus* Infections
**TNF-α**	Macrophages T-cells	Proinflammatory	Aggravate *S. aureus* induced septic arthritis but protective in sepsis [118].
**IL-1**	Macrophages Dendritic cells Endothelial cells	Proinflammatory	Protective in *S. aureus* induced septic arthritis and sepsis [119].
**IL-6**	Macrophages and T cells	Proinflammatory	Elevated IL-6 levels in synovial fluid from septic arthritis patients [120]. The role of IL-6 was not yet assessed in animal models for septic arthritis and sepsis.
**IL-12**	Monocytes Macrophages Dendritic cells	Proinflammatory	Protective in *S. aureus* induced sepsis but not septic arthritis [121]
**IL-4**	Th2 cells	Anti-inflammatory	Dual role in *S. aureus* induced septic arthritis and sepsis depending on the genetic background of the host [122,123].
**IL-10**	Monocytes Dendritic cells T-cells	Anti-inflammatory	Protective in *S. aureus* induced septic arthritis [124].
**IL-17**	Th17 cells	Proinflammatory	Protective in local but not systemic *S. aureus* infection [114].
**IFN-γ**	NK cells T-cells	Proinflammatory	Protective in *S. aureus* induced sepsis but aggravates septic arthritis [125].
**Janus kinase**	All cells	Proinflammatory	Protective in *S. aureus* septic arthritis but pathogenic in sepsis [126]

## Data Availability

Not applicable.

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
