# Peer review of "Bacteria and Host Interplay in Staphylococcus aureus Septic Arthritis and Sepsis"

_pathogens, 2021, doi:10.3390/pathogens10020158_

Round 1

Reviewer 1 Report

In this review the authors clearly summarize and provide new insights about the interplay between bacteria and host in determining Staphylococcus aureus septic arthritis and sepsis. In addition to review the recent literature concerning septic arthritis and sepsis caused by Staphylococcus aureus in the context of the interplay beetwen bacteria and host, the authors identify fundamental questions about this subject to be answered in the future. The authors  have an in-depth knowledge of the argument as is evident from the numerous articles published on this topic. The manuscript is nicely presented and well written.

Author Response

Dear Reviewer,

Many thanks for nice words.

Best wishes, Tao Jin

Reviewer 2 Report

This reveiw by Tao Jin et al. describe very interesting data on the bacteria/host interplay in S. aureus septic arhtritis and sepsis.

Numerous modifications are needed.

At first, a logical plan, in respect with section/subsection organization in author's recommendation could be of interest to ease the understanding by the reader. For example, the theoretical part on S. aureus pathophysiology and structure has to be at first in the manuscript, ending it with BJI epidemiology and treatment.

All acronym have to be fully explicited before their first use. All bacterial, gene name have to be indicated in italic. All numbers below twelve have to be written in full letters. Beware IL-17A is underlined (line 545)

"S. aureus and its virulence factors" paragraphs would benefit from table summarizing the content of each parts. In this part of the manuscript, "bacterial DNA" paragraph is pretty short and deserve to be detailled.

Paragraph on IL-4 (lines 522-527) is very interesting and have to be detailed in order to understand the discrepant results of the impact of this cytokine.

Finally, this review lack 3 major parts to be considered as complete. 1/ the authors have to sumarized all therapies that focused on host/bacteria interaction in SA BJI infections; 2/ as gram positive bacteria are major producers of biofilms that are key for antibiotic choice, the authors have to precise its production/impact/regulation in BJI.;  3/ interplay between host and SA could not be fully understood without giving data on the studies focusing on microbiome characterization in this context.

Author Response

At first, a logical plan, in respect with section/subsection organization in author's recommendation could be of interest to ease the understanding by the reader. For example, the theoretical part on S. aureus pathophysiology and structure has to be at first in the manuscript, ending it with BJI epidemiology and treatment.

: We are grateful for this constructive comment. The structure of the manuscript is changed as reviewer suggested. We start with S. aureus virulence factors and host immune responses, followed by disease epidemiology and treatments.

All acronym have to be fully explicited before their first use. All bacterial, gene name have to be indicated in italic. All numbers below twelve have to be written in full letters. Beware IL-17A is underlined (line 545)

: We have changed all matters according to your suggestion. 

"S. aureus and its virulence factors" paragraphs would benefit from table summarizing the content of each parts. In this part of the manuscript, "bacterial DNA" paragraph is pretty short and deserve to be detailled.

: Fully agree. A table regarding the S. aureus and its virulence factors is now added. Bacterial DNA part is now described in more details.

Paragraph on IL-4 (lines 522-527) is very interesting and have to be detailed in order to understand the discrepant results of the impact of this cytokine.

: More discussion is now added to IL-4 section.

Finally, this review lack 3 major parts to be considered as complete. 1/ the authors have to sumarized all therapies that focused on host/bacteria interaction in SA BJI infections; 2/ as gram positive bacteria are major producers of biofilms that are key for antibiotic choice, the authors have to precise its production/impact/regulation in BJI.; 3/ interplay between host and SA could not be fully understood without giving data on the studies focusing on microbiome characterization in this context.

: Fully agree. Now we have added three sections including all therapies that focused on host/bacteria interaction, biofilm infections, and microbiome characterization. 

Round 2

Reviewer 2 Report

After this extensive revision, the manuscript is now suitable for publication. I would like to thank the authors for their very complete work regarding to my comments.

Author Response

Thank you so much for the comments. We do appreciate your constructive comments.

Best wishes, Tao Jin